# Cognitive and Speech Rehabilitation in a Patient Affected by Takotsubo Cardiomyophathy: A Case Report

**DOI:** 10.3390/medicina58060697

**Published:** 2022-05-24

**Authors:** Francesco Corallo, Lilla Bonanno, Caterina Formica, Valentina Coppola, Marcella Di Cara, Piercataldo D’aleo, Silvia Marino, Chiara Smorto, Viviana Lo Buono

**Affiliations:** IRCCS Centro Neurolesi Bonino Pulejo, 98124 Messina, Italy; francesco.corallo@irccsme.it (F.C.); katia.formica@irccsme.it (C.F.); coppolavalentina1994@gmail.com (V.C.); marcella.dicara@irccsme.it (M.D.C.); piercataldo.daleo@irccsme.it (P.D.); silvia.marino@irccsme.it (S.M.); chiara.smorto@irccsme.it (C.S.); viviana.lobuono@irccsme.it (V.L.B.)

**Keywords:** cognitive rehabilitation, ischemic cerebral event, neuropsychology, takotsubo syndrome

## Abstract

*Background and Objectives*: Takotsubo Syndrome (TS) constitutes one of the most recent clinical realities in modern cardiology. It is clinically similar to the acute coronary syndrome, in the absence of obstructive coronary artery disease. *Case Presentation*: We described a case of a female patient affected by TS and left ventricular apical thrombus. Several studies described the cardiological syndrome, overlooking the neuropsychological and psychological outcomes. We aimed to assess the advantages of an integrated, multidisciplinary and multifunctional rehabilitation. *Conclusions*: This specific training contributed to reducing the tolerance to frustration given by her communication’s difficulty. It has favored a good therapeutic alliance and a good success of the psychotherapeutic path, guaranteeing the reduction of her anxious symptoms and an improvement in the emotive and relational status.

## 1. Introduction

Takotsubo Syndrome (TS) constitutes one of the most recent clinical realities in modern cardiology. It is clinically similar to acute coronary syndrome, in the absence of obstructive coronary artery disease [1]. The acute left ventricular dysfunction is a known potential mechanism in cardiac embolism disease [2], which in turn is an important risk factor for strokes [2,3,4]. The syndrome is also known as “stress-induced cardiomyopathy” [5] or “broken heart syndrome” [6]. TS represents one of the strongest pieces of evidence that stress is a powerful risk factor for ischemic heart attack, even in the absence of abnormalities in the coronary circulation [7]. Recent studies have demonstrated that TS is associated with cardiovascular complications including stroke [8].

Stroke in TS is thought to be secondary to left ventricular mural thromboembolism explained by blood stasis from wall motion abnormalities and hypercoagulability from catecholamine surge. Although TS is not primarily an atherosclerotic disease, several studies have demonstrated patients with TS to have a high prevalence of atherosclerotic risk factors.

Despite the disabling motor and cognitive outcomes, to our knowledge there is no standardized rehabilitative protocol for stroke TS. Therefore, we considered the implementation of a multidisciplinary rehabilitative protocol to be of preeminent importance. For this reason, we described the outcomes and subsequent stroke in a 61-year-old female patient with a stress-induced heart attack.

## 2. Case Report

We report the case of a 61-year-old aphasic female that came to our neurorehabilitative center after an ischemic cerebral event. She presented a severe medical history; in fact, she arrived with a diagnosis of TS and left ventricular apical thrombus, which subsequently developed into a stroke event.

The clinical criteria were the modified Mayo Clinic criteria for the diagnosis of TS [9]: (i) akinesia or dyskinesia of left ventricular segments with regional wall motion abnormalities that extended beyond the distribution of a single epicardial vessel, (ii) absence of obstructive coronary artery disease, (iii) new electrocardiographic abnormalities, and (iv) absence of pheochromocytoma or myocarditis.

She has been treated with oral anticoagulants and antiplatelet pharmacological therapies. The patient was on anticoagulation before the ischemic stroke. The anticoagulant therapy was continued in the acute phase and during her neurorehabilitation treatment. The patient underwent a full cardiological evaluation and hematological exams, which confirmed the TS diagnosis. The patient reported a cognitive decline with particular deficits in the language domains; moreover, she showed emotional disorders (anxiety and depression) (Table 1).

The patient performed a 3Tesla Magnetic Resonance imaging (MRI) examination, which showed an extended hyperintense area in the fronto-temporal-insular left lobe, due to an ischemic attack in the acute/hyperacute phase (Figure 1).

This brain insult resulted in cognitive and motor outcomes. In particular, the patient showed Broca’s aphasia and right hemiplegia. The complexity of this case required the implementation of a multidisciplinary intervention. The patient was subjected every day to physiotherapeutic training to improve motor deficits, speech therapy to improve her language difficulties, and neuropsychological intervention to teach an alternative strategy for communication. Motor training was based on task-oriented exercises; balance exercises and gait training have been carried out, as they improve mobility. The cognitive training included specific exercises, paper-and-pencil or computerized tasks oriented toward attention/working memory, psychomotor speed, executive functions, visuo-spatial abilities, and calculation skills. Speech language training included an intensive stimulation of the speech intensity, exercises to improve verbal intelligibility, and treatment of limitations to swallowing, choking and slowness of chewing. Every day, a rehabilitative program was organized according to a predefined scheme: motor exercises (60 min), language exercises (60 min) and cognitive exercises (60 min). The rehabilitation program lasted eight weeks. Rehabilitation also included systemic-relational therapy conducted by a psychotherapist. The latter is a form of psychotherapy that focuses on how an individual’s personal relationships, behavioral patterns, and life choices are interconnected with the issues they face in their life. In this case, a psychotherapy intervention aimed at mitigating marked anxious and depressive symptoms, facilitating a better awareness of her new health condition, and favoring the acquisition of adaptive strategies for social reintegration.

After approximately three months of neurocognitive and speech therapy, she showed great improvement in her capacity for repetition, denomination and comprehension.

## 3. Discussion

Takotsubo cardiomyopathy is not a rare complication of acute ischemic stroke. It occurs most often soon after stroke onset and is commonly asymptomatic. Females predominantly constitute the stroke patients who have experienced the development of TS.

We described the case of an aphasic female who suffered from an ischemic stroke followed by stress-induced cardiomyopathy. Several studies describe this cardiological syndrome, overlooking the neuropsychological and psychological outcomes. To the best of our knowledge, this is the first described case of neuropsychological and psychotherapeutic intervention.

Our strategy aimed at exploiting the advantages of an integrated, multidisciplinary, and multifunctional rehabilitation. This specific training contributed to reducing her tolerance to frustration given by her communication difficulty. It has favored a good therapeutic alliance and the success of the psychotherapeutic path, guaranteeing a reduction of her anxious symptoms and improving her emotive and relational status. The view according to which psychotherapy is efficacious in people with depression after stroke is supported by studies on patients with a range of acquired brain injuries [10] and those with other neurological conditions. Specifically, systemic-relational therapy seems to be a useful method of support in the rehabilitation of patients with strokes, favoring the process of accepting the new conditions of life within a network of relationships [11].

This integrated method seems to be efficient in guaranteeing a better level of daily life functioning, making the most of cognitive and adaptation resources. Moreover, this case proves that stress is a powerful risk factor for ischemic heart attack, even in the absence of abnormalities in the coronary circulation [7]. Stressful events are widely experienced in everyone’s life and represent a trigger of Takotsubo syndrome in susceptible patients [2].

We were able to discover that our patient’s whole life has involved highly stressful and challenging events, like an unlucky marriage, the sudden disappearance of her husband, the discovery of an illegitimate son from another woman, and the severe mental disease of her daughter. It is important to act on the predisposition factors in a timely manner and to try to eliminate the risk and stressor factors.

## Figures and Tables

**Figure 1 medicina-58-00697-f001:**
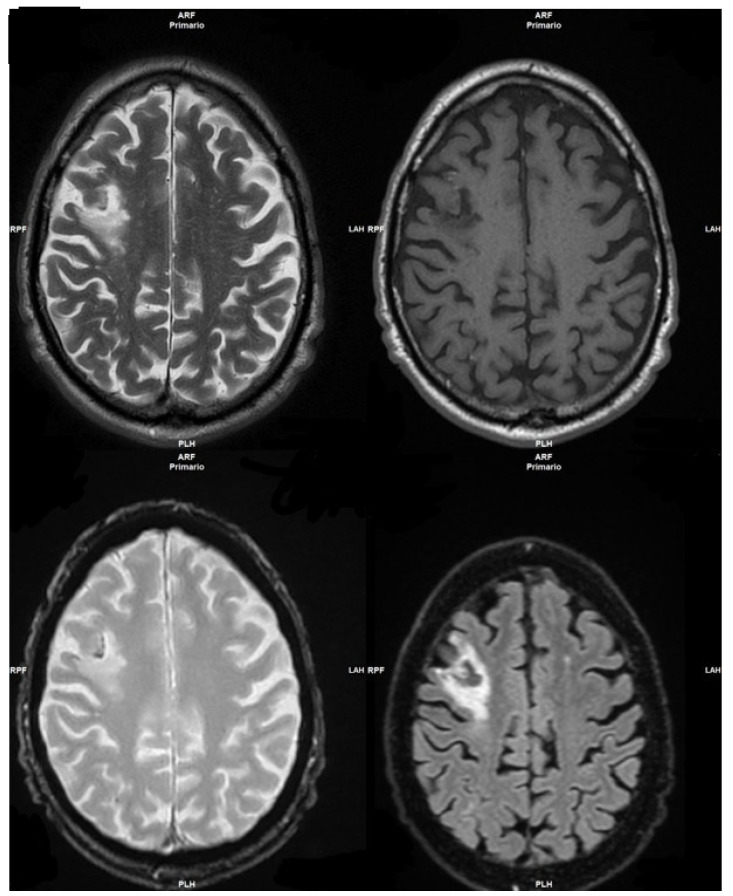
Hyperintense area in the fronto-temporal-insular left lobe.

**Table 1 medicina-58-00697-t001:** Neuropsychological assessment.

Neuropsychological Assessment	Score T1	Score T2
* **Aachener Aphasie Test (AAT)** *		
Spontaneous language	0	2
Token Testing	47	26
Repetition	0	74
Written language	0	10
Designation	0	66
Understanding	68	100
*Clinical Scales*		
* **Functional Independence Measure (FIM)** *	39	73
* **Misure of Anxiety** *		
Hamilton Rating Scale (HAM)	18	10
* **Misure of Depression** *		
Beck Depression Inventory (BDI-II)	29	10

## Data Availability

Not applicable.

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
