# Peer review of "Cognitive and Speech Rehabilitation in a Patient Affected by Takotsubo Cardiomyophathy: A Case Report"

_medicina, 2022, doi:10.3390/medicina58060697_

Round 1

Reviewer 1 Report

The authors describe the case of a patient with cardioembolic stroke following takotsubo cardiomyopathy, highlighting the importance of a multidisciplinary rehabilitation  to mitigate neurological and psychological consequences. While the subject is of clinical significance, however, a multidisciplinary and multifunctional approach like the one presented in the article should follow every stroke whether takotsubo related or not. 

  • The authors are advised to elaborate on clinical details concerning takotsubo cardiomyopathy and the clinical course of stroke, presenting the timeline preferably in a table. 
  • Further information concerning the multidisciplinary rehabilitation protocol is needed.
  • Administered medical therapy should be added in the case presentation. Please clarify whether the patient was on anticoagulation before the ischemic stroke. Was anticoagulant therapy continued in the acute phase, and if not when was it initiated?
  • Has the protocol suggested by the authors been used in other patients with stroke? Was it effective?
  • The authors should recheck the conclusion section in the abstract.
  • Language needs extensive editing. 

Reviewer 2 Report

Dear Authors

The presented case report is interesting, but requires serious corrections.

  1. Abstract - the fragment is an instruction for the authors.
  2. Introduction, lines 37-38 - Takotsubo syndrome is not a heart attack, it is clinically similar to acute coronary syndrome.
  3. What happened first in the patient? Takotsubo or stroke?
  4. The greater part of the "Case Report" on psychotherapy should be included in the discussion, while the "Case Report" should be supplemented with data on the course of the disease, diagnosis and treatment.
  5. Case report - how was the patient diagnosed with Takotsubo syndrome? Based on what criteria? What was the stress factor? He proposes to complete more clinical data on Takotsubo (patient's clinical symptoms, changes in ECG, troponin, NT-proBNP, coronary angiography, etc.). What anticoagulant and antiplatelet medications were she receiving? What other cardiological, neurological or psychiatric medications did the patient receive?
  6. I propose to supplement the description with pictures of the left ventricle from coronary angiography or ECHO examination of the heart.
  7. Was a follow-up echocardiographic examination performed?
  8. I propose to add more details concerning the applied psychotherapy. Was the psychotherapy used in the patient the standard psychotherapy used in patients after cerebral stroke? Has psychotherapy been shown to be beneficial in patients in other cardiovascular diseases?

Best regards

Round 2

Reviewer 2 Report

Dear Authors

"Authors 'responses to reviewers' comments" has not been included.

Some of the changes proposed in the review were introduced in the text, but not all of them. There is no answer to the rest.

I do not see any possibility to complete the review.

Additional remarks:

  • currently the correct valid name is Takotsubo Syndrome (TS)
  • in the abstract, please delete the word "Results"
  • the manuscript requires an editorial correction

Regards.
